# Non-Circular Signal DOA Estimation with Nested Array via Off-Grid Sparse Bayesian Learning

**DOI:** 10.3390/s23218907

**Published:** 2023-11-01

**Authors:** Xudong Dong, Jun Zhao, Meng Sun, Xiaofei Zhang

**Affiliations:** 1College of Electronic and Information Engineering, Nanjing University of Aeronautics and Astronautics, Nanjing 211106, China; nanhangdxd@nuaa.edu.cn (X.D.); zhangxiaofei@nuaa.edu.cn (X.Z.); 2College of Electronic and Information Engineering, Tongji University, Shanghai 201804, China

**Keywords:** off-grid sparse Bayesian learning, DOA estimation, non-circular signal, nested array

## Abstract

For the traditional uniform linear array (ULA) direction of arrival (DOA) estimation method with a limited array aperture, a non-circular signal off-grid sparse Bayesian DOA estimation method based on nested arrays is proposed. Firstly, the extended matrix of the received data is constructed by taking advantage of the fact that the statistical properties of non-circular signals are not rotationally invariant. Secondly, we use the difference and sum co-arrays for the nested array technique, thus increasing the array aperture and improving the estimation accuracy. Finally, we take the noise as part of the interest signal and iteratively update the grid points using the sparse Bayesian learning (SBL) method to eliminate the modeling errors caused by off-grid gaps. The simulation results show that the proposed algorithm can improve the accuracy of DOA estimation compared with the existing algorithms.

## 1. Introduction

Direction of arrival (DOA) estimation is a problem in the field of array signal processing that has been widely used in radar, sonar, indoor positioning, and mobile communications [1], as well as in some near-field localization scenarios [2,3]. The earliest subspace-based super-resolution DOA estimation methods were developed, including multiple signal classification (MUSIC) [4], estimation of signal parameters via rotational invariance techniques (ESPRIT) [5], weighted subspace fitting (WSF) [6], etc. Usually, these algorithms use the uniform linear array (ULA) to receive the signal, where the *M*-element ULA can only distinguish M−1 sources at most. Many non-uniform arrays [7], such as minimum redundant array (MRA) [8], co-prime array (CA) [9,10,11], and nested array (NA) [12,13,14,15], can increase the degrees of freedom (DOFs) and improve the accuracy of DOA estimation by employing the virtualization technology. However, the virtualization technique turns the received signal into a single snapshot vector, resulting in rank loss of the covariance matrix. Scholars propose many effective methods for decoherence, such as spatial smoothing (SS)-based MUSIC (SS-MUSIC [12]) and Toeplitz-based algorithms [16]. Nevertheless, SS-MUSIC and Toeplitz-based algorithms seriously destroy the DOFs, so they require a large number of snapshots to obtain the signal or noise subspace accurately.

In recent years, algorithms based on sparse representation (SR) [17], compressive sensing (CS) [18,19], and sparse Bayesian learning (SBL) [20] have resolved coherent sources with limited snapshots and have gradually become hotspots in DOA estimation research. The $l_{1}$-SVD [21] algorithm assumes that the unknown source is on the grid, which can be a sparse signal representation (SSR) problem when the source of interest constitutes a sparse signal condition. An off-grid DOA estimation model based on CS is proposed in [22], where the estimated DOA no longer depends on a fixed grid. Yang Z et al. propose the off-grid sparse Bayesian inference (OGSBI) method [23] and developed an iterative algorithm by dividing the search range uniformly into grids, maintaining high estimation accuracy, even with very coarse sampling grids. Subsequently, Zhang proposed an improved off-grid SBL method to reduce the effect of noise variance [24], but a significant problem with this method is that the performance depends on the trade-off between accuracy and computational effort. In [25], the author proposes a computationally efficient SBL method for root off-grid DOA estimation that takes the sampling locations within the coarse grid as adjustable parameters and refines the coarse grid using the expectation–maximization (EM) algorithm [26], significantly reducing computational complexity and eliminating model errors. Although the above algorithms perform well in the off-grid field, the estimation accuracy of the algorithms can be further improved if combined with sparse arrays.

In [27], the author proposes a received model for a nested array that treats the noise variance as part of the unknown signal of interest and iterates the grid points by SBL, eliminating the model errors caused by off-grid gaps. In [28], the second-order Taylor expansion is used to replace the first-order Taylor expansion to alleviate the gap error from the grid, but it increases the amount of calculation. Since the nested array technique [27,28] constitutes a virtual differential co-array that is uniformly distributed and has no holes, the DOFs can be fully utilized.

However, the methods mentioned above only pay attention to the difference co-array of sparse arrays while ignoring the contribution of sum co-arrays. Non-circular (NC) signals [29,30] (such as binary phase shift keying (BPSK), pulse amplitude modulation (PAM), and amplitude shift keying (ASK)) can expand the array aperture and increase the DOFs, which are widely used in modern digital modulation schemes such as telecommunication or satellite systems [31,32,33]. The author in [34] converts the received model with NC signals into a real-valued sparse model and proposes a real-valued signal processing method in impulsive noise based on SBL. In [35], the author proposes a new SBL method to solve the off-grid DOA estimation problem, which can automatically identify sources from a grid of candidate angles and match the direction information of interest from covariance and pseudo-covariance vectors when circular and NC signals coexist. In [36], a new ESPRIT-like method is proposed, which derives a generalized covariance and is suitable for strictly NC signals. Nonetheless, the above-mentioned methods [34,35,36] are based on ULA, so the application of NC signals with sparse arrays may need to be discussed.

Through the above analysis, we propose an NC signal DOA estimation with the nested array method based on off-grid sparse Bayesian learning. The main contributions are as follows:We combine virtual difference co-arrays and sum co-arrays by exploiting the property of NC signals, which extends the array aperture and improves estimation accuracy.We take the noise variance as part of the NC signals of interest and then iterate over the internal parameters by the OGSBI method to maintain the standard SBL form after computing the selection matrix and removing redundant information in the nested array.

Other parts of this paper are organized as follows. Some backgrounds are presented in Section 2, including the data model and the concepts of sum and difference co-arrays. The mathematical analysis and discussion of the DOA estimation of the proposed algorithm is provided in Section 3. Numerical simulation and the conclusion are presented in Section 4 and Section 5, respectively.


**Statement** **1:**
*uppercase (lowercase) bold characters are used to represent matrices (vectors). The ${\left(\cdot \right)}^{T}$, ${\left(\cdot \right)}^{H}$, and ${\left(\cdot \right)}^{*}$ operators correspond to the transpose, conjugate transpose, and conjugate operations, respectively. The $diag\left(\cdot \right)$ notation is employed for diagonal matrices, while vec$\left(\cdot \right)$ represents the vectorization operation. $\langle a,b\rangle$ indicates the set $\left\{x\in Z\left|a⩽x⩽b\right.\right\}$, the symbols ⊗, ⊙ and ⊕ stand for the Kronecker, Khatri-Rao, and Hadamard products, respectively. Matrix $I_{N}$ denotes an identity matrix of size N×N, and E signifies the expectation operator. Finally, $\left|\cdot \right|$ is used to represent the magnitude of a complex value, and Re· denotes the real part of the complex variable.*




**Statement** **2:**
*We collect all the acronyms employed throughout the manuscript in Table 1 for the readers’ convenience.*



## 2. Background

### 2.1. The Data Model

Consider *K* narrowband far-field sources $\theta_{k},k=1,2,\cdots ,K$ impinging on a nested array that consists of two concatenated ULAs (shown in Figure 1), where the inner and outer ULAs have *M* and *N* sensors with array spacing *d* and $\left(M+1\right)d$, respectively. $d=\frac{\lambda }{2}$ and λ is the signal carrier wavelength. The positions of the sensors can be expressed as L=Ld, where
(1)$$\begin{array}{c} L=\left\{l_{i},i=1,2,\cdots ,P\right\}\\ =\left\{0,1,\cdots ,M-1,M,2(M+1)-1,\cdots ,N(M+1)-1\right\}, \end{array}$$
is a set containing the location information of all sensors, and $l_{1}<l_{2}<\cdots <l_{i}<\cdots <l_{P}$, P=M+N denotes the number of sensors.

We consider *K* far-field narrowband independent sources $s_{k}\left(t\right),k=1,\cdots ,K$, t=1,2,⋯,T impinging into the linear sparse array, where *T* denotes the number of snapshots. Then, the received data at time *t* can be indicated as
(2)$$y\left(t\right)=\sum_{k=1}^{K}a\left(\theta_{k}\right)s_{k}\left(t\right)+n\left(t\right)=\mathrm{As}\left(t\right)+n\left(t\right),$$
where $s\left(t\right)={\left[s_{1}\left(t\right),s_{2}\left(t\right),\cdots ,s_{K}\left(t\right)\right]}^{T}\in C^{K\times 1}$ is the signal vector and the noise $n\left(t\right)$ obeys a Gaussian distribution, i.e., $n\left(t\right)∼CN\left(\cdot \left|0,\sigma_{n}^{2}\right.\right)$, $\sigma_{n}^{2}$ denotes the noise variance and it is uncorrelated with the signal. $A=\left[a\left(\theta_{1}\right),\cdots ,a\left(\theta_{k}\right),\cdots ,a\left(\theta_{K}\right)\right]\in C^{P\times K}$ is the direction matrix with
(3)$$a\left(\theta_{k}\right)={\left[1,e^{-j\frac{2\pi }{\lambda }l_{2}d\;\sin\theta_{k}},\cdots ,e^{-j\frac{2\pi }{\lambda }l_{P}d\;\sin\theta_{k}}\right]}^{T}.$$

Considering the strict NC signal model [37], the received data Equation (Equation 2) can be rewritten as
(4)$$y\left(t\right)=A\Phi s_{R}\left(t\right)+n\left(t\right).$$
where $\Phi =\mathrm{diag}\left\{e^{-j\varphi_{1}},\cdots ,e^{-j\varphi_{k}},\cdots ,e^{-j\varphi_{K}}\right\}$ with $\varphi_{k}$ the NC phase of the *k*-th circular signal $s_{R}\left(t\right)$, and $s_{R}\left(t\right)\in R^{K\times 1}$. By exploiting the NC characteristics of the signal, the received signal can be extended as
(5)$$\begin{array}{c} y_{0}\left(t\right)=\left[\begin{array}{c} y\left(t\right)\\ y^{*}\left(t\right) \end{array}\right]=\left[\begin{array}{c} A\Phi \\ A^{*}\Phi^{*} \end{array}\right]s_{R}\left(t\right)+\left[\begin{array}{c} n\left(t\right)\\ n^{*}\left(t\right) \end{array}\right]\\ =Es_{R}\left(t\right)+n_{0}\left(t\right), \end{array}$$
where $E=\left[e\left(\theta_{1},\varphi_{1}\right),\cdots ,e\left(\theta_{K},\varphi_{K}\right)\right]\in C^{2P\times K}$ is the extended direction matrix with $e\left(\theta_{k},\varphi_{k}\right)=\left[\begin{array}{c} a\left(\theta_{k}\right)e^{-j\varphi_{k}}\\ a^{*}\left(\theta_{k}\right)e^{j\varphi_{k}} \end{array}\right]$ the *k*-th extended steering vector and $n_{0}\left(t\right)$ represents the extended noise. Then, the covariance matrix of the output matrix $y_{0}\left(t\right)$ can be
(6)$$R_{y}=E\left\{y_{0}\left(t\right)y_{0}^{H}\left(t\right)\right\}=ER_{S}E^{H}+\sigma_{n}^{2}I_{2\left(M+N\right)},$$
where $R_{S}=E\left\{s_{R}\left(t\right){s_{R}}^{H}\left(t\right)\right\}=diag\left\{\delta_{1}^{2},\delta_{2}^{2},\cdots ,\delta_{K}^{2}\right\}$ is the signal covariance matrix, where E denotes the expectation operator, and $I_{2\left(M+N\right)}$ is the identity matrix. By ‘vectorizing’ $R_{y}$, we obtain
(7)$$y≜vec\left(R_{y}\right)=\left(E^{*}⊙E\right)p+\sigma_{n}^{2}1_{n},$$
where $E^{*}⊙E=\left[\dot{e}\left(\theta_{1},\varphi_{1}\right),\cdots ,\dot{e}\left(\theta_{k},\varphi_{k}\right),\cdots ,\dot{e}\left(\theta_{K},\varphi_{K}\right)\right]$ with $\dot{e}\left(\theta_{k},\varphi_{k}\right)=e^{*}\left(\theta_{k},\varphi_{k}\right)⊗e\left(\theta_{k},\varphi_{k}\right)$ is the *k*-th virtual array steering vector, $p={\left[\delta_{1}^{2},\delta_{2}^{2},\cdots ,\delta_{K}^{2}\right]}^{T}$ is a single snapshot signal vector, $1_{n}={\left[i_{1}^{T},\cdots ,i_{l}^{T},\cdots ,i_{2\left(M+N\right)}^{T}\right]}^{T}$ with $i_{l}^{T}\in R^{2P\times 1},l=1,2,\cdots ,2P$ being a column vector of all zeros except 1 at the *l*-th position. The covariance matrix $R_{y}$ is usually estimated from finite snapshots, i.e.,
(8)$${\hat{R}}_{y}\approx \frac{1}{T}\sum_{t=1}^{T}y_{0}\left(t\right){y_{0}}^{H}\left(t\right).$$

According to Equations (Equation 6) and (Equation 8), there exists an approximation error [38,39]
(9)$$\varepsilon ≜vec\left({\hat{R}}_{y}-R_{y}\right)∼CN\left(0,\frac{1}{T}R_{y}^{T}⊗R_{y}\right).$$Then, Equation (Equation 7) can be approximated as
(10)$$\hat{y}≜vec\left({\hat{R}}_{y}\right)=\left(E^{*}⊙E\right)p+\sigma_{n}^{2}1_{n}+\varepsilon .$$

### 2.2. Difference and Sum Co-Arrays

The vectorization process corresponds to a sum-and-difference operation on the position information in the array steering vectors. The difference co-array is given by
(11)$$D=\left\{l_{i}-l_{j},l_{i},l_{j}\in L\right\},$$
and the positive and negative sum co-arrays are given as follows: [40]
(12)$$\begin{array}{c} S_{+}=\left\{l_{i}+l_{j},l_{i},l_{j}\in L\right\},\\ S_{-}=\left\{-l_{i}-l_{j},l_{i},l_{j}\in L\right\}. \end{array}$$

According to [12], the virtual difference co-array formed by the nested array is a uniform linear array with a range of $\langle -\left(MN+N-1\right),MN+N-1\rangle$. For the positive sum co-array, the range of the ULA is $\langle 0,MN+M+N-1\rangle$, while the negative sum co-array with ULA ranges from $-\left(MN+M+N-1\right)$ to 0. By introducing a virtual array of sum co-arrays and difference co-arrays, the continuous ULA can be denoted as
(13)$$\begin{array}{c} \langle -\left(MN+M+N-1\right),0\rangle \cup \\ \langle -\left(MN+N-1\right),MN+N-1\rangle \cup \langle 0,MN+M+N-1\rangle . \end{array}$$

For example, when M=3, N=3. Figure 2 shows that the difference co-array is $\langle -11,11\rangle d$, the continuous negative sum co-array is $\langle -14,0\rangle d$, the continuous positive sum co-array is $\langle 0,14\rangle d$, and the ULA can be expressed as
(14)$$\langle -14,0\rangle d\cup \langle -11,11\rangle d\cup \langle 0,14\rangle d.$$

## 3. The Proposed Method

### 3.1. Data Extension

From Equation (Equation 10), we have
(15)$$E_{0}=E^{*}⊙E=\left[\begin{array}{c} A^{*}\Phi^{*}\\ A\Phi \end{array}\right]⊙\left[\begin{array}{c} A\Phi \\ A^{*}\Phi^{*} \end{array}\right],$$
where $E_{0}=\left[\dot{e}\left(\theta_{1},\varphi_{1}\right),\cdots ,\dot{e}\left(\theta_{k},\varphi_{k}\right),\cdots ,\dot{e}\left(\theta_{K},\varphi_{K}\right)\right]$, and
(16)$$\dot{e}\left(\theta_{k},\varphi_{k}\right)=\left[\begin{array}{c} a^{*}\left(\theta_{k}\right)e^{j\varphi_{k}}\\ a\left(\theta_{k}\right)e^{-j\varphi_{k}} \end{array}\right]⊗\left[\begin{array}{c} a\left(\theta_{k}\right)e^{-j\varphi_{k}}\\ a^{*}\left(\theta_{k}\right)e^{j\varphi_{k}} \end{array}\right].$$

For ease of computation, we introduce a row exchange matrix $J\in R^{4P^{2}\times 4P^{2}}$ [40], i.e.,
(17)$$J=\left[\begin{array}{cc} \left[\begin{array}{c} J_{1}\\ J_{2} \end{array}\right] & 0_{2P^{2}}\\ 0_{2P^{2}} & \left[\begin{array}{c} J_{1}\\ J_{2} \end{array}\right] \end{array}\right],$$
where $J_{1}=I_{P}⊗\left[\begin{array}{cc} I_{P} & 0_{P} \end{array}\right]\in R^{P^{2}\times 2P^{2}}$, and $J_{2}=I_{P}⊗\left[\begin{array}{cc} 0_{P} & I_{P} \end{array}\right]\in R^{P^{2}\times 2P^{2}}$. Through the row exchange matrix, the extended direction vector becomes
(18)$$\begin{array}{c} c\left(\theta_{k},\varphi_{k}\right)=J\dot{e}\left(\theta_{k},\varphi_{k}\right)=\\ \left[\begin{array}{c} a^{*}\left(\theta_{k}\right)⊗a\left(\theta_{k}\right)\\ a^{*}\left(\theta_{k}\right)⊗a^{*}\left(\theta_{k}\right)e^{2j\varphi_{k}}\\ a\left(\theta_{k}\right)⊗a\left(\theta_{k}\right)e^{-2j\varphi_{k}}\\ a\left(\theta_{k}\right)⊗a^{*}\left(\theta_{k}\right) \end{array}\right]=\left[\begin{array}{c} c_{1}\\ c_{2}\\ c_{3}\\ c_{4} \end{array}\right], \end{array}$$
where $c_{1}=c_{4}^{*}$ is the steering vector of the difference co-array, $c_{2}$ denotes the steering vector of the negative sum co-array, and $c_{3}$ is the steering vector of the positive sum co-array. Then, $J\hat{y}$ can be expressed as
(19)$$\begin{array}{c} z=J\hat{y}=JE_{0}p+J\sigma_{n}^{2}1_{n}+J\varepsilon \\ \;=\mathrm{Cp}+\sigma_{n}^{2}{1^{\prime }}_{n}+\tilde{\varepsilon }, \end{array}$$
where $C=JE_{0}=\left[c\left(\theta_{1},\varphi_{1}\right),\cdots ,c\left(\theta_{k},\varphi_{k}\right),\cdots ,c\left(\theta_{K},\varphi_{K}\right)\right]$ means a direction matrix of virtual difference co-arrays and sum co-arrays, $\tilde{\varepsilon }=J\varepsilon$, and ${1^{\prime }}_{n}=J1_{n}$.

### 3.2. Sparse Bayesian Inference for DOA Estimation

The received data in Equation (Equation 2) can be solved by a sparse approximation method [21]. Let $\Theta =\left[{\hat{\theta }}_{1},{\hat{\theta }}_{2},\cdots ,{\hat{\theta }}_{I}\right]$, where *I* represents the number of sampling grids, which satisfies I≫K and $r={\hat{\theta }}_{I}-{\hat{\theta }}_{I-1}$ is the grid resolution. As [27], the over-complete basis can be written as
(20)$$\hat{z}={\bar{C}}_{0}\bar{p}+\sigma_{n}^{2}{1^{\prime }}_{n}+\bar{\varepsilon }=\left[\begin{array}{cc} {\bar{C}}_{0} & {1^{\prime }}_{n} \end{array}\right]\left[\begin{array}{c} \bar{p}\\ \sigma_{n}^{2} \end{array}\right]+\bar{\varepsilon },$$
where ${\bar{C}}_{0}=\left[c\left(\theta_{1},\varphi_{1}\right),\cdots ,c\left(\theta_{I_{k}},\varphi_{I_{k}}\right),\cdots ,c\left(\theta_{I},\varphi_{I}\right)\right]$ with $c\left(\theta_{I_{k}},\varphi_{I_{k}}\right)=\;[a^{*}\left(\theta_{I_{k}}\right)⊗a\left(\theta_{I_{k}}\right);$$a^{*}\left(\theta_{I_{k}}\right)⊗a^{*}\left(\theta_{I_{k}}\right)e^{2j\varphi_{I_{k}}};\;a\left(\theta_{I_{k}}\right)⊗a\left(\theta_{I_{k}}\right)e^{-2j\varphi_{I_{k}}};a\left(\theta_{I_{k}}\right)⊗a^{*}\left(\theta_{I_{k}}\right)]$, $\bar{p}$ is a zero-padded extension matrix of p whose non-zero elements correspond to the true DOA at $\left\{\theta_{k},k=1,\cdots ,K\right\}$, and $\bar{\varepsilon }∼CN\left(0,W\right),W=\frac{J\left(R_{Y}^{T}⊗R_{Y}\right)J^{T}}{T}$.

Ideally, DOA is assumed to be on the grid. However, in practical scenarios, this phenomenon often only exists sometimes. To deal with the problem of the grid-gap, a method based on the linear approximation of the first-order Taylor is proposed in [23]. Assuming that ${\hat{\theta }}_{I_{k}},I_{k}\in \left\{1,2,\cdots ,I\right\}$ is the nearest grid point to a DOA $\theta_{k}\notin \Theta$, the steering vector can be linearized as
(21)$$c\left(\theta_{k},\varphi_{k}\right)\approx c\left({\hat{\theta }}_{I_{k}},\varphi_{I_{k}}\right)+b\left({\hat{\theta }}_{I_{k}},\varphi_{I_{k}}\right)\left(\theta_{k}-{\hat{\theta }}_{I_{k}}\right),$$
where $b\left({\hat{\theta }}_{I_{k}},\varphi_{I_{k}}\right)=c^{\prime }\left({\hat{\theta }}_{I_{k}},\varphi_{I_{k}}\right)$ is the first-order derivative of $c\left({\hat{\theta }}_{I_{k}},\varphi_{k}\right)$ to ${\hat{\theta }}_{I_{k}}$. Equation (Equation 20) can be rewritten as
(22)$$\hat{z}=\left[\begin{array}{cc} {\bar{C}}_{0}\left(\beta \right) & {1^{\prime }}_{n} \end{array}\right]\left[\begin{array}{c} \bar{p}\\ \sigma_{n}^{2} \end{array}\right]+\bar{\varepsilon },$$
where $B=\left[b\left({\hat{\theta }}_{1},\varphi_{1}\right),\cdots ,b\left({\hat{\theta }}_{I_{k}},\varphi_{I_{k}}\right),\cdots ,b\left({\hat{\theta }}_{I},\varphi_{I}\right)\right]$ and ${\bar{C}}_{0}\left(\beta \right)={\bar{C}}_{0}+Bdiag\left\{\beta \right\}$, $\beta ={\left[\beta_{1},\beta_{2},\cdots ,\beta_{I}\right]}^{T}$ is a zero vector except that the $I_{k}$-th element ${\left[\beta \right]}_{I_{k}}=\theta_{k}-{\hat{\theta }}_{I_{k}},$
k=1,2,⋯,K.

According to Equation (Equation 18), $\hat{z}$ can be divided into four parts, i.e., $\hat{z}=\left[\begin{array}{cccc} z_{1}; & z_{2}; & z_{3}; & z_{4} \end{array}\right]$. Since there are holes in sum co-arrays, we only select continuous virtual array elements, i.e., $S_{-}^{\prime }=\langle -\left(MN+M+N-1\right),0\rangle$ and $S_{+}^{\prime }=\langle 0,MN+M+N-1\rangle$. We remove the redundancy items on $z_{1},z_{2},z_{3},z_{4}$, which depend on select matrices $F_{1}$, $F_{2}$, $F_{3}$, and $F_{4}$. We define the weight functions for the virtual array locations D, $S_{-}^{\prime }$ and $S_{+}^{\prime }$ as
(23)$$\begin{array}{c} \omega_{1}=\langle \left\{\mathrm{card}\left(\mu_{1}\right)|\mu_{1}=l_{i}-l_{j},l_{i},l_{j}\in L,\mu_{1}\in D\right\}\rangle \\ \omega_{2}=\langle \left\{\mathrm{card}\left(\mu_{2}\right)|\mu_{2}=-l_{i}-l_{j},l_{i},l_{j}\in L,\mu_{2}\in {S^{\prime }}_{-}\right\}\rangle \\ \omega_{3}=\langle \left\{\mathrm{card}\left(\mu_{3}\right)|\mu_{3}=l_{i}+l_{j},l_{i},l_{j}\in L,\mu_{3}\in {S^{\prime }}_{+}\right\}\rangle \\ \omega_{4}=\langle \left\{\mathrm{card}\left(\mu_{4}\right)|\mu_{4}=l_{i}-l_{j},l_{i},l_{j}\in L,\mu_{4}\in D\right\}\rangle \end{array}$$
where $\mathrm{card}\left(\mu \right)$ is the number of elements μ. Let $D_{v}=MN+N-1,S_{v}=MN+M+N-1$, i,j=1,2,⋯,P, then the corresponding selection matrix can be expressed as follows:(24)$$\begin{array}{c} {\left[F_{1}\right]}_{\mu_{1}+D_{v}+1,i+P\left(j-1\right)}=\left\{\begin{array}{c} \frac{1}{\omega_{1}\left(\mu_{1}\right)},\mu_{1}=l_{i}-l_{j}\\ 0,otherwise \end{array}\right.\\ {\left[F_{2}\right]}_{\mu_{2}+S_{v}+1,i+P\left(j-1\right)}=\left\{\begin{array}{c} \frac{1}{\omega_{2}\left(\mu_{2}\right)},\mu_{2}=-l_{i}-l_{j}\\ 0,otherwise \end{array}\right.\\ {\left[F_{3}\right]}_{\mu_{3}+1,i+P\left(j-1\right)}=\left\{\begin{array}{c} \frac{1}{\omega_{3}\left(\mu_{3}\right)},\mu_{3}=l_{i}+l_{j}\\ 0,otherwise \end{array}\right.\\ {\left[F_{4}\right]}_{\mu_{4}+D_{v}+1,i+P\left(j-1\right)}=\left\{\begin{array}{c} \frac{1}{\omega_{4}\left(\mu_{4}\right)},\mu_{4}=l_{j}-l_{i}\\ 0,otherwise \end{array}\right.. \end{array}$$

By removing the redundancy items of $\hat{z}$, we can obtain $\tilde{z}$, where ${\tilde{z}}_{1}={\tilde{z}}_{4}^{*}\in C^{(2D_{v}+1)\times 1}$ represents the difference co-array data. ${\tilde{z}}_{2}={\tilde{z}}_{3}^{*}\in C^{\left(S_{v}+1\right)\times 1}$, ${\tilde{z}}_{2}$ denotes the negative sum co-array data while ${\tilde{z}}_{3}$ represents the positive sum co-array data. Hence, all valid received data can be expressed as $\tilde{z}=\left[{\tilde{z}}_{1};{\tilde{z}}_{2};{\tilde{z}}_{3};{\tilde{z}}_{4}\right]\in C^{(4D_{v}+2S_{v}+4)\times 1}$, i.e.,
(25)$$\begin{array}{c} \tilde{z}=F\hat{z}\\ \;=\left[\begin{array}{cccc} F_{1} & & & \\ & F_{2} & & \\ & & F_{3} & \\ & & & F_{4} \end{array}\right]\left[\begin{array}{c} z_{1}\\ z_{2}\\ z_{3}\\ z_{4} \end{array}\right]=\left[\begin{array}{c} {\tilde{z}}_{1}\\ {\tilde{z}}_{2}\\ {\tilde{z}}_{3}\\ {\tilde{z}}_{4} \end{array}\right]. \end{array}$$

According to Equations (Equation 22) and (Equation 25), we have
(26)$$\tilde{z}=F\underset{\Psi \left(\beta \right)}{\underset{︸}{\left[\begin{array}{cc} {\bar{C}}_{0}\left(\beta \right) & {1^{\prime }}_{n} \end{array}\right]}}\underset{d}{\underset{︸}{\left[\begin{array}{c} \bar{p}\\ \sigma_{n}^{2} \end{array}\right]}}+F\tilde{\varepsilon }=F\Psi \left(\beta \right)d+\vec{\varepsilon },$$
where d represents the sparse signal, and $\vec{\varepsilon }$, after removing the redundancy, satisfies a complex Gaussian distribution, i.e.,
(27)$$\vec{\varepsilon }=F\cdot \left[W^{-\frac{1}{2}}\left[\hat{z}-\Psi \left(\beta \right)d\right]\right]∼CN\left(0,I_{\left(4D_{v}+2S_{v}+4\right)}\right).$$

A typical SBL treatment of d involves assigning a non-stationary Gaussian prior distribution with variance $\delta_{i}$ to each d element. Suppose that hyperparameter $\Lambda =diag\left(\delta \right)$, and $p\left(d|\delta \right)=CN\left(d|0,\Lambda \right)$ with $\delta ={\left[\delta_{1},\cdots ,\delta_{i},\cdots ,\delta_{I+1}\right]}^{T}$, where δ can be modeled as a Gamma distribution [23], i.e.,
(28)$$\Gamma \left(\delta \right)=\prod_{i=1}^{I+1}\Gamma \left(\delta_{i};1,\upsilon \right),$$
where υ is a small positive constraint (e.g., υ = 0.01 [23,41]). From Equation (Equation 26), we have
(29)$$p\left(\tilde{z}|d,\beta \right)=CN\left(\tilde{z}|F\Psi \left(\beta \right)d,Q\right),$$
where $Q={\mathrm{FWF}}^{T}$. Assuming that d is a hidden variable, the posterior probability density [23,26] is
(30)$$p\left(d|\tilde{z},\delta ,\beta \right)=CN\left(d|\mu ,\Sigma \right),$$
where
(31)$$\mu =\Sigma \left(\Psi^{H}\left(\beta \right)F^{T}\right)Q^{-1}\tilde{z},$$
(32)$$\Sigma ={\left(\Psi^{H}\left(\beta \right)F^{T}Q^{-1}F\Psi \left(\beta \right)+\Lambda^{-1}\right)}^{-1}.$$

Then, the EM algorithm [23] can be applied for recursive calculating until it reaches a prescribed accuracy. In the E-step, we need to have the lower bound of $\ln p\left(\tilde{z},\delta ,\beta \right)$, which is
(33)$$\begin{array}{c} L\left(\delta ;\beta \right)=E{\left\{\ln p\left(d,\tilde{z},\delta ,\beta \right)\right\}}_{p\left(d|\tilde{z},\delta ,\beta \right)}\\ =E{\left\{\ln p\left(\tilde{z}|d,\beta \right)p\left(d|\delta \right)p\left(\delta \right)\right\}}_{p\left(d|\tilde{z},\delta ,\beta \right)}, \end{array}$$
where $E{\left\{\cdot \right\}}_{p\left(x\right)}$ represents the expected value of $p\left(x\right)$. In the M-step, the hyperparameter updates that maximize the lower bound function
(34)$$\left(\delta^{new},\beta^{new}\right)=\arg\max_{\delta ,\beta }L\left(\delta ;\beta \right).$$

According to [27], the hyperparameter updates for $\delta_{i}$ can be simplified as
(35)$$\delta_{i}^{new}=\frac{-1+\sqrt{1+4\upsilon {\left[\mu \mu^{H}+diag\left(\Sigma \right)\right]}_{ii}}}{2\upsilon },i=1,2,\cdots ,I+1,$$
where ${\left[\cdot \right]}_{ii}$ denotes the $\left(i,i\right)$-th element of the matrix. However, since the structure of the construction matrix $F\Psi \left(\beta \right)$ is different in [23], we need to redefine the update of $\beta^{new}$.

### 3.3. Grid Refining

Ignoring the independent terms in Equation (Equation 33), we obtain
(36)$$\begin{array}{c} E{\left\{\ln p\left(\tilde{z}|d,\beta \right)\right\}}_{p\left(d|\tilde{z},\delta ,\beta \right)}\\ =-E{\left\{{\left(\tilde{z}-F\Psi \left(\beta \right)d\right)}^{H}Q^{-1}\left(\tilde{z}-F\Psi \left(\beta \right)d\right)\right\}}_{p\left(d|\tilde{z},\delta ,\beta \right)}\\ =-E{\left\{{∥Q^{-1/2}\cdot \left(\tilde{z}-F\Psi \left(\beta \right)d\right)∥}_{2}^{2}\right\}}_{p\left(d|\tilde{z},\delta ,\beta \right)}\\ =-\underset{a}{\underset{︸}{{∥Q^{-1/2}\tilde{z}-Q^{-1/2}F\Psi \left(\beta \right)\mu ∥}_{2}^{2}}}-\underset{b}{\underset{︸}{tr\left(Q^{-1}F\Psi \left(\beta \right)\Sigma \Psi^{H}\left(\beta \right)F^{T}\right)}}+const. \end{array}$$

The term *a* in the above formula can be simplified to Equation (Equation 37),
(37)Q−1/2z˜−Q−1/2FΨβμ22=Q−1/2z˜︸Zw−Q−1/2FC¯0(β)μI−μ0Q−1/2F1′n︸Iw22=Q−1/2z˜︸Zw−Q−1/2FC¯0︸Cw+Q−1/2FB︸BwdiagβμI−μ0Q−1/2F1′n︸1′w22=Zw−CwμI−BwdiagβμI−μ01′w22=βTBwTBw*⊕μIμIHβ−2RediagμI*BwHZw−CwμI−μ0IwTβ+const
where $\mu_{I}$ represents the sub-vector of μ, whose elements are indexed from 1 to *I*, and $\mu_{0}$ represents the end element of μ, i.e., $\mu ={\left[\begin{array}{cccc} \mu_{1} & \cdots & \mu_{I} & \mu_{0} \end{array}\right]}^{T}={\left[\begin{array}{cc} \mu_{I} & \mu_{0} \end{array}\right]}^{T}$. Let $\Sigma =\left[\begin{array}{cc} \Upsilon & \gamma \\ \gamma^{H} & \gamma_{0} \end{array}\right]$.

Similarly, term *b* can be rewritten as Equation (Equation 38).
(38)trQ−1FΨβΣΨHβFT=trFΨβH·Q−1/2HQ−1/2FΨβΣ=trQ−1/2FC¯0+FBdiagβ,F1′nHQ−1/2FC¯0+FBdiagβ,F1′nΣ=trCw+Bwdiagβ,1′wHCw+Bwdiagβ,1′wΣ=trCw+Bwdiagβ,1′wHCwΥ+BwdiagβΥ+1′wγH,Cwγ+Bwdiagβγ+1′wγ0=trCw+BwdiagβHCwΥ+BwdiagβΥ+1′wγH+1′wHBwdiagβγ+const=2RetrBwHCwΥdiagβ+trdiagβΥdiagβBwHBw+trdiagβBwH1′wγH+1′wHBwdiagβγ+const=2RediagBwHCwΥT+1′wHBwdiagγβ+βTΥ⊕BwTBw*β+const

Substituting Equations (Equation 37) and (Equation 38) into Equation (Equation 36), we have
(39)$$\begin{array}{c} E{\left\{\ln p\left(\tilde{z}|d,\beta \right)\right\}}_{p\left(d|\bar{Z},\delta ,\beta \right)}\\ =-\beta^{T}P\beta +2v^{T}\beta +const \end{array}$$
with
(40)P=ReBwTBw*⊕μIμIH+Υ,
(41)v=RediagμI*BwHZw−CwμI−μ01′w−RediagBwHCwΥ+diagγBwT1′w*.

Find the partial derivative to β in Equation (Equation 39), then
(42)$$\beta =P^{-1}v.$$

If P is invertible, Equation (Equation 42) holds. Otherwise, $\beta_{i}=\frac{v_{i}}{P_{ii}},\forall i$. Reference [27] proposes a grid update method to reduce the interval error from the grid. Similarly, we define the grid to update as follows:(43)$${\hat{\theta }}_{I_{k}}^{new}=\left\{\begin{array}{c} {\hat{\theta }}_{I_{k}},{\hat{\theta }}_{I_{k}}+\beta_{I_{k}}\notin \left[{\hat{\theta }}_{I_{k}-1},{\hat{\theta }}_{I_{k}+1}\right]\\ {\hat{\theta }}_{I_{k}}+\beta_{I_{k}},{\hat{\theta }}_{I_{k}}+\beta_{I_{k}}\in \left[{\hat{\theta }}_{I_{k}-1},{\hat{\theta }}_{I_{k}+1}\right] \end{array}\right..$$

Then, the $\Psi \left(\beta \right)$ matrix can provide a better approximation of the actual steering matrix; we update the $\Psi \left(\beta \right)$ matrix with the new grid ${\hat{\theta }}_{I_{k}}^{new}$ so that the next iteration is closer to the true value, i.e., $\Psi \left(\beta \right)=\left[\begin{array}{cc} {\bar{C}}_{0}+B\mathrm{diag}\left(\beta^{new}\right) & {1^{\prime }}_{n} \end{array}\right]$. Table 2 illustrates the pseudocode of the proposed DOA estimation algorithm.

## 4. Numerical Simulation

To evaluate the performance of the proposed algorithm, we compare the proposed algorithm with the SS-MUSIC method [12], the $l_{1}$-SVD method [17], the OGSBI-ULA method [23], and the OGSBI-NA method [27]. The Cramér–Rao bound (CRB) for circular and NC signals [42] is considered for comparison. To compare the performance of the proposed algorithm with those reported recently, we assume that the NC phases of all signals are consistent $\varphi_{1}=\varphi_{2}=\cdots =\varphi_{K}=5^{\circ }$. $\delta_{k}^{2}$ is the power of the *k*-th signal, and $\sigma_{n}^{2}$ is the noise power, and the signal-to-noise ratio $\mathrm{SNR}=10\log_{10}\left(\delta_{k}^{2}/\sigma_{n}^{2}\right)$. We employ the root mean square error (RMSE) to evaluate the performance of DOA estimation, which is defined as
(44)$$\mathrm{RMSE}=\sqrt{\frac{1}{\mathrm{MC}K}\left(\sum_{i=1}^{\mathrm{MC}}\sum_{k=1}^{K}{\left({\hat{\theta }}_{k,i}-\theta_{k}\right)}^{2}\right)},$$
where ${\hat{\theta }}_{k,i}$ is the estimation value of the *k*-th signal in the *i*-th Monte Carlo (MC), and $\theta_{k}$ represents the true DOA of the *k*-th signal.

### 4.1. Computational Complexity

In this section, we use the number of multiplications of real (or complex) numbers as a criterion for the complexity. The pseudocode of the proposed algorithm is given in Table 2, and the computational complexity of the proposed algorithm consists of many components, such as the nested array non-circular technique, the OGSBI algorithm, and the grid refining operation. In order to facilitate the reader’s understanding, we provide the main computational complexities of the proposed algorithms (computational expressions and the corresponding computational complexity) in Table 3. In addition, the computational complexity of the proposed algorithm is mainly increased by the non-circular extension of the signal compared to the OGSBI-NA algorithm, but the metric level is still $O\left\{P^{2}\right\}$.

### 4.2. The Spatial Spectrum with Different DOAs

In the first simulation, the spatial spectrum of the proposed method and OGSBI-NA [27] are estimated with different values of DOA. The following different DOAs are considered:

(a) $\theta_{k}\in \left[\begin{array}{cc} -50^{\circ }+\eta , & 30^{\circ }+\eta \end{array}\right]$ with $\eta =10^{\circ }\times \mathrm{randn}\left(-1,1\right)$, *K* = 2;

(b) $\theta_{k}\in \left[-50^{\circ }+\eta ,-30^{\circ }+\eta ,30^{\circ }+\eta ,50^{\circ }+\eta \right]$ with $\eta =10^{\circ }\times \mathrm{randn}\left(-1,1\right)$, *K* = 4;

(c) $\theta_{k}\in \left[-50^{\circ }:10^{\circ }:50^{\circ }\right]+\eta$ with $\eta =10^{\circ }\times \mathrm{randn}\left(-1,1\right)$, *K* = 11;

(d) $\theta_{k}\in \left[-70^{\circ }:10^{\circ }:70^{\circ }\right]+\eta$ with $\eta =10^{\circ }\times \mathrm{randn}\left(-1,1\right)$, *K* = 15.

Figure 3 illustrates the spatial spectrum of the proposed algorithm and OGSBI-NA when SNR = 0 dB, the number of snapshots *T* = 200, and the grid resolution $2^{\circ }$. In Figure 3a,b, there are fewer sources (*K* = 2 or *K* = 4) than the number of array elements (P=10). Both the OGSBI-NA and the proposed method successfully find all sources, but the OGSBI-NA signal peaks have lower magnitudes than the proposed algorithm. In Figure 3c–d, there are more sources (*K* = 11 or *K* = 15) than the number of array sensors (P=10). Obviously, both the OGSBI-NA method and the proposed algorithm in this paper successfully locate all 11 or 15 sources. However, the amplitudes of some signal peaks of the OGSBI-NA method are lower than those of the proposed algorithm, which indicates that our method is superior to the existing algorithms.

### 4.3. The RMSE of Overdetermined DOA Estimation

We consider 10 physical array elements, i.e., *M* = 5, *N* = 5. There are two narrowband NC signals $\left[\begin{array}{cc} -50^{\circ }+\eta , & 30^{\circ }+\eta \end{array}\right]$ with NC phases $\varphi =5^{\circ }$ impinging on the nested array, where $\eta =10^{\circ }*\mathrm{randn}\left(-1,1\right)$ to eliminate the prior information that may be contained in the predefined DOA set.

Figure 4 compares the RMSE performance and SNR with *T* = 200. As shown in Figure 4, since the grid-gap is relatively large (the grid resolution $2^{\circ }$), the $l_{1}$-SVD method and the SS-MUSIC method have similar accuracy. Based on a nested array, the proposed method has higher performance than the OGSBI-NA method because of the use of conjugate information formed by NC signals.

The RMSE results vs. the number of snapshots with SNR = 0 dB are depicted in Figure 5. As the number of snapshots increases, the DOA estimation accuracies of $l_{1}$-SVD and SS-MUSIC improve slightly. However, these two methods perform worse than the proposed algorithm, mainly due to the relatively large grid-gap. In addition, the accuracy of the OGSBI-ULA and OGSBI-NA methods also increases with the increase in the number of snapshots. When the number of snapshots reaches 500, the RMSE of the OGSBI-NA algorithm drops to 0.05, indicating that using nested arrays can improve the estimation accuracy. With the increase in the number of snapshots, the RMSE of the proposed algorithm is the lowest. At the same time, it is proved that the conjugate property of NC signals could improve the estimation accuracy.

### 4.4. The RMSE of Underdetermined DOA Estimation

To demonstrate that the proposed algorithm can estimate more sources than the number of physical sensors, we investigate the performance of the proposed algorithm in underdetermined DOA estimation scenarios. For this purpose, we assume 11 narrowband NC signals with NC phases $\varphi =5^{\circ }$ from directions $\left[-50^{\circ }:10:50^{\circ }\right]+\eta$ incident to a nested array of *M* = *N* = 5, $\eta =10^{\circ }*\mathrm{randn}\left(-1,1\right)$ and the grid resolution is $2^{\circ }$. When the number of sources is greater than the number of array elements, the $l_{1}$-SVD method and the OGSBI-ULA method do not work, so we compare the proposed algorithm with the SS-MUSIC method [12] and the OGSBI-NA method [27].

We then choose the SNR of each signal to vary from −4 dB to 10 dB, *T* = 200, MC = 200. The RMSEs of three methods are given, including SS-MUSIC, OGSBI-NA (the grid resolution is $2^{\circ }$), and the proposed algorithm. As shown in Figure 6, SS-MUSIC still has a large error when the number of estimated sources exceeds the number of array elements. The performances of the three algorithms gradually improve as the SNR increases. Compared to the OGSBI-NA method, the RMSE performance of the algorithm proposed in this problem is superior, thanks to the use of the conjugate information created by the NC signal, which extends the virtual array aperture.

Then, we kept the SNR at 0 dB, and increased the snapshot number from 50 to 1000, and MC = 200. The RMSEs of the three methods vs. the number of snapshots are shown in Figure 7. The accuracy of DOA estimation by SS-MUSIC slightly improves as the number of snapshots increases, but the error is still relatively large due to the low signal-to-noise ratio. When the snapshots T>300, the RMSE of the proposed algorithm is close to that of OGSBI-NA. As the number of snapshots increases, the RMSE of the algorithm gradually stabilizes. It is demonstrated that the conjugate property of NC signals can improve the accuracy of estimation.

## 5. Conclusions

In this paper, a DOA estimation algorithm based on off-grid sparse Bayesian inference using nested arrays with NC signals is proposed. Since the statistical properties of NC signals do not have rotational invariance, the received data can be expanded. Based on this advantage, we construct difference co-arrays and sum co-arrays, which increase the array aperture and improve the estimation accuracy of DOA. Then, we consider the noise as part of the signal of interest and use it for the recursion of the OGSBI method after computing the selection matrix and removing redundant information. After iterations of sparse Bayesian inference, we can update the grid and find the optimal value. The simulation results show that the proposed algorithm significantly enhances the accuracy of DOA estimation. Future research could focus on developing advanced techniques for processing NC signals. It may involve exploring new signal processing methods or statistical models to better capture the unique properties of NC signals and avoid the limitation of considering only the same NC phases.

## Figures and Tables

**Figure 1 sensors-23-08907-f001:**
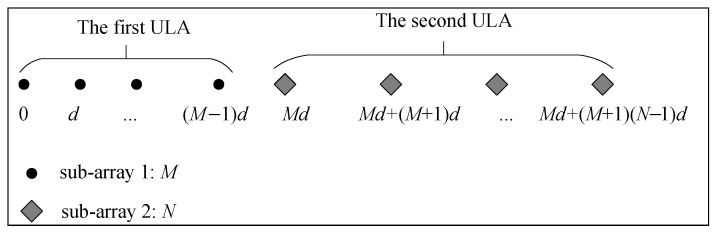
The nested array model.

**Figure 2 sensors-23-08907-f002:**
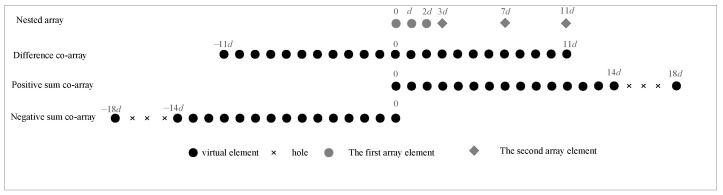
Difference and sum co-arrays, where M=3, N=3.

**Figure 3 sensors-23-08907-f003:**
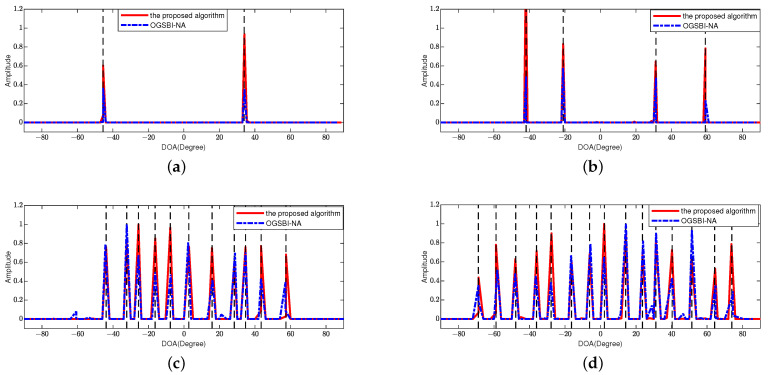
Amplitude with different methods, *T* = 200, SNR = 0 dB, (**a**) K=2; (**b**) K=4; (**c**) K=11; (**d**) K=15.

**Figure 4 sensors-23-08907-f004:**
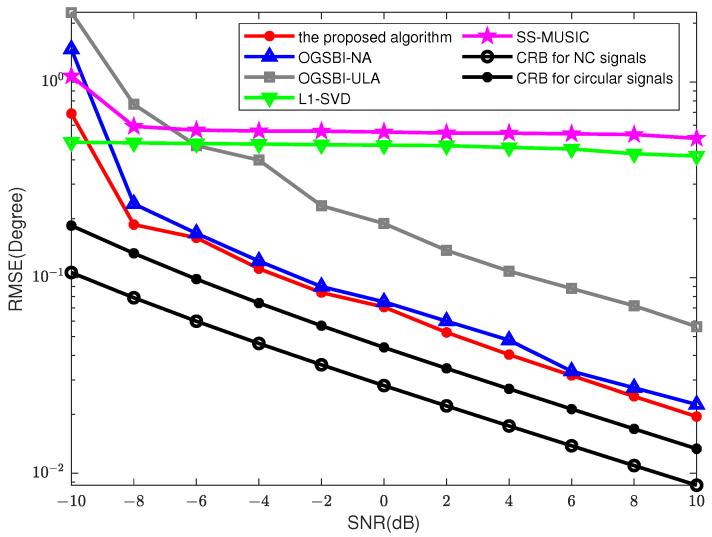
RMSEs between the DOA estimation and different SNRs, *T* = 200, MC = 200.

**Figure 5 sensors-23-08907-f005:**
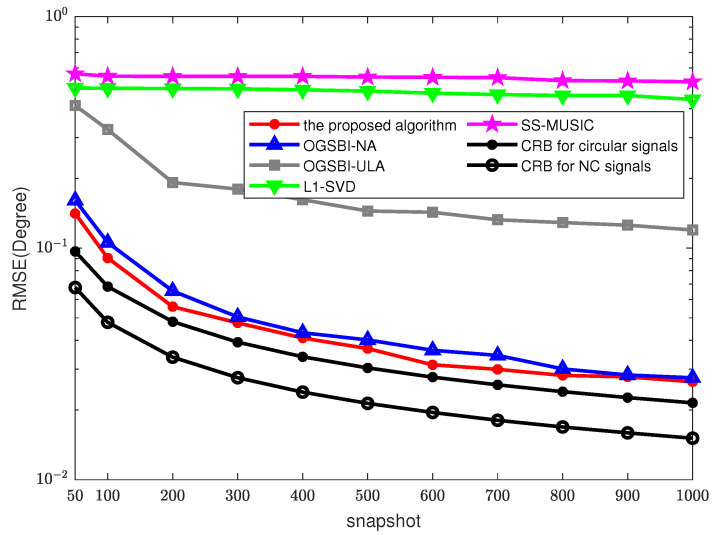
RMSEs with different snapshots, SNR = 0 dB, MC = 200.

**Figure 6 sensors-23-08907-f006:**
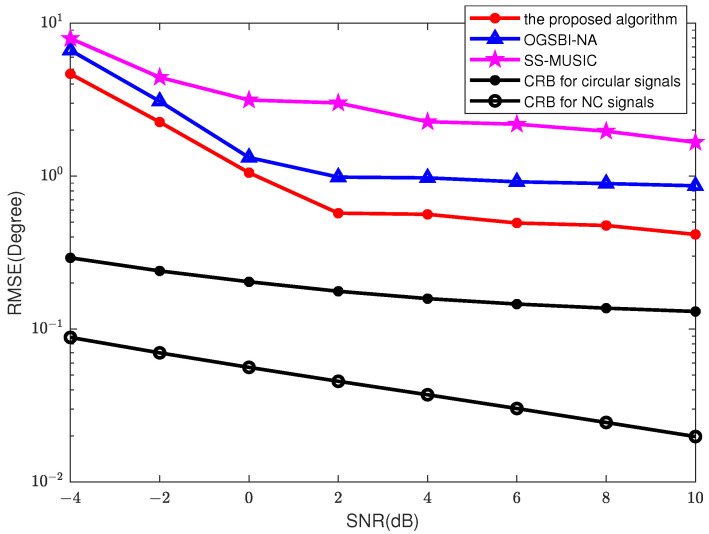
RMSEs for different SNRs, the snapshots *T* = 200, MC = 200.

**Figure 7 sensors-23-08907-f007:**
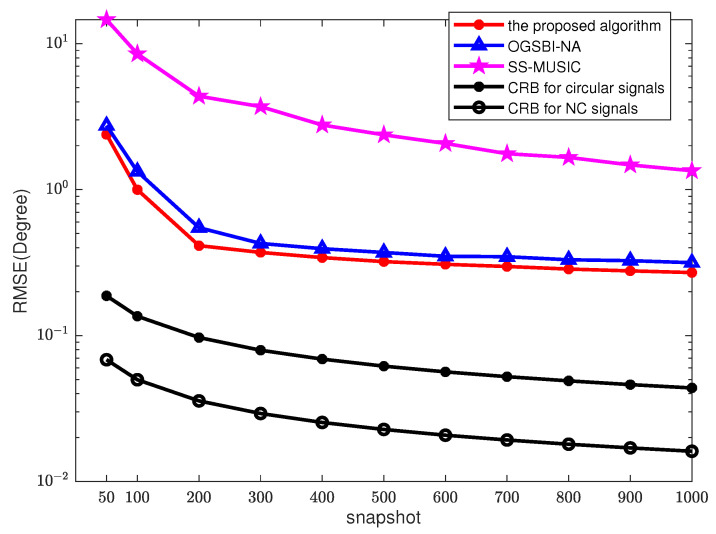
RMSEs vs. different snapshots, SNR = 0 dB, MC = 200.

**Table 1 sensors-23-08907-t001:** Acronyms and their full names.

Acronyms	Full Name
DOA	direction of arrival
ULA	uniform linear array
SBL	sparse Bayesian learning
MUSIC	multiple signal classification
ESPRIT	estimation of signal parameters via rotational invariance techniques
WSF	weighted subspace fitting
CA	co-prime array
NA	nested array
MRA	minimum redundant array
DOFs	degrees of freedom
SS-MUSIC	spatial smoothing MUSIC
CS	compressive sensing
SR	sparse representation
SSR	sparse signal representation
OGSBI	off-grid sparse Bayesian inference
EM	expectation–maximization
NC	non-circular
PAM	pulse amplitude modulation
BPSK	binary phase shift keying
ASK	amplitude shift keying
CRB	Cramér–Rao bound
RMSE	root mean square error

**Table 2 sensors-23-08907-t002:** The pseudocode of the proposed algorithm.

Input: $y_{0}\left(t\right),t=1,2,\cdots ,t$,${\bar{C}}_{0}\left(\beta \right)$ and ${1^{\prime }}_{n}$.
**Output**: Parameter estimator: $\delta^{new}$, $\beta^{new}$ and ${\hat{\theta }}_{I_{k}}^{new}$.
**1: Initialization**: Set $\Psi \left(\beta \right)\;=\;\left[\begin{array}{cc} {\bar{C}}_{0}\left(\beta \right) & {1^{\prime }}_{n} \end{array}\right]$, $\beta_{0}=1$ and $\delta_{0}=0_{I+1}$.
**2:** Calculate the covariance matrix: ${\hat{R}}_{Y}=\frac{1}{T}\sum_{t=1}^{T}y_{0}\left(t\right){y_{0}}^{H}\left(t\right)$.
**3:** Vectorize ${\hat{R}}_{Y}$, obtain $\hat{y}$ according to Equation (Equation 10). Then, multiply the row
exchange matrix J to obtain Equation (Equation 19).
**4:** Construct the over-complete information $\hat{z}$ according to Equation (Equation 22).
**5:** Remove the redundancy items according to Equation (Equation 25), we can obtain $\tilde{z}$.
**6:** Calculate the weight matrix W, use it to normalize the vectorized covariance matrix, and then go through the remove redundancy matrix F to obtain Equation (Equation 27).
**7:** Build d and $\Psi \left(\beta \right)$ based on the current values of δ and β separately.
**8: While**$\frac{{∥\delta_{t}-\delta_{t-1}∥}_{2}}{{∥\delta_{t}∥}_{2}}⩽10^{-3}$, **do**
Calculate the mean μ and covariance Σ according to Equation (Equation 31) and Equation (Equation 32), respectively.
Update the δ according to Equation (Equation 35), respectively.
Calculate the P and v.
Update the β according to Equation (Equation 42).
Update the grid ${\hat{\theta }}_{I_{k}}^{new}$ according to Equation (Equation 43).
Update $\Psi \left(\beta \right)$.
**9: end**

**Table 3 sensors-23-08907-t003:** Computational expressions and computational complexity.

Computational Expressions	Computational Complexity
$Q=\mathrm{FW}F^{T}$	$O\left\{4P^{2}G\left(4P^{2}+G\right)\right\}$, $G=4D_{v}+2S_{v}+4$
$Q^{-1}$	$O\left\{G^{3}\right\}$
$\mu =\Sigma \left(\Psi^{H}\left(\beta \right)F^{T}\right)Q^{-1}\tilde{z}$, Equation (Equation 31)	$O\left\{4P^{2}\left(I_{1}^{2}+I_{1}G\right)+I_{1}G+I_{1}G^{2}\right\}$,$I_{1}=I+1$
$\Lambda^{-1}$	$O\left\{I_{1}^{3}\right\}$
$\Sigma ={\left(\Psi^{H}\left(\beta \right)F^{T}Q^{-1}F\Psi \left(\beta \right)+\Lambda^{-1}\right)}^{-1}$, Equation (Equation 32)	$O\left\{4P^{2}\left(2I_{1}G+I_{1}G^{2}+I_{1}^{2}\right)+2I_{1}^{3}\right\}$
δ, Equation (Equation 35)	$O\left\{I_{1}^{2}\right\}$
$B_{w}=Q^{-1/2}\mathrm{FB}$	$O\left\{4P^{2}\left(G^{2}+IG\right)+G^{3}\right\}$
$Q^{-1/2}F{\bar{C}}_{0}$	$O\left\{4P^{2}\left(G^{2}+IG\right)+G^{3}\right\}$
$Q^{-1/2}F{1^{\prime }}_{n}$	$O\left\{4P^{2}\left(G^{2}+G\right)+G^{3}\right\}$
P=ReBwTBw*⊕μIμIH+Υ, Equation (Equation 40)	$O\left\{I^{2}G+2I^{2}\right\}$
v, Equation (Equation 41)	$O\left\{3I^{2}G+3IG+I^{3}\right\}$

## Data Availability

The data used in this paper can be requested from the corresponding authors upon request.

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
