# Peer review of "Non-Circular Signal DOA Estimation with Nested Array via Off-Grid Sparse Bayesian Learning"

_sensors, 2023, doi:10.3390/s23218907_

Round 1
Reviewer 1 Report
Comments and Suggestions for Authors
Please review the pasted file.

English writing needs to be improved.
Reviewer 2 Report
Comments and Suggestions for Authors
1) At the beginning of Sec. I, please recall the need for the design of DoA and localization approaches also in the context of near-field conditions:
Devaney, Anthony J. "Time reversal imaging of obscured targets from multistatic data." IEEE Transactions on Antennas and Propagation 53.5 (2005): 1600-1610.
Ciuonzo, D. (2017). On time-reversal imaging by statistical testing. IEEE Signal Processing Letters, 24(7), 1024-1028.
2) Please collect all the acronyms employed throughout the manuscript in a given table for readers’ convenience.
3) Please add a mathematical notation paragraph at the end of Sec. I.
4) Organization paragraph in Sec. I, there is a typo: “Numerical simulation and conclusion are given in sections 4 and ??, respectively.”
5) Before Eq. (7): “By vectoring” -> “By vectorizing”
6) Please discuss the computational complexity involved in the proposed algorithm by using the well-known Big O notation.
7) Sec. 4- Please briefly recall the computational complexity involved in the considered baselines, as well as please briefly describe their working principle.
8) Conclusion section should be enriched with a brief paragraph describing future research directions.
Mostly OK
Reviewer 3 Report
Comments and Suggestions for Authors
Please see the attached file.

Grammar and writing style need to be improved.
Reviewer 4 Report
Comments and Suggestions for Authors
This paper proposes a DOA estimation algorithm using nested arrays, rotationally variant signals, and with sparse Bayesian learning.
None of these methods is novel. However, the combination of these methods may have some level of novelty.
Simulations results show that the proposed algorithm outperforms some existing algorithms. The contribution may be sufficient.
Summarizing, I recommend a minor revision.
- Please highlight the main differences with existing algorithms, such as [23], whose first part shares several similarities with the first part of this submission.
- Line 84: The cross-reference to the Section "Conclusion" is missing, please correct it.
- Line between (8) and (9): There is an English mistake, please correct it.
- Please check the whole paper for other possible typos.
Round 2
Reviewer 2 Report
Comments and Suggestions for Authors
The authors have satisfactorily addressed my previous concerns.
Reviewer 3 Report
Comments and Suggestions for Authors
Previous comments have been addressed.
Comments on the Quality of English LanguageGrammar and writing style can be further improved.